# Population Reinforcement of the Endangered Freshwater Pearl Mussel (*Margaritifera margaritifera*): Lessons Learned

Louise Lavictoire [1,*] and Christopher West [2]

1   Freshwater Biological Association, The Ferry Landing, Far Sawrey, Ambleside LA22 0LP, UK
2   West Cumbria Rivers Trust, Convention Centre, Skiddaw St., Keswick CA12 4BY, UK; chris@westcumbriariverstrust.org
*   Correspondence: llavictoire@fba.org.uk

**Abstract:** Freshwater mussel populations are in sharp decline and are considered to be one of the most imperilled groups globally. Consequently, the number of captive breeding programmes has increased rapidly in recent years, coupled with subsequent reintroductions/population reinforcements to reverse these declines. The outcomes of mussel conservation translocations are seldom reported in the primary literature, hindering opportunities for learning and for population recovery at pace. Here, we describe the methods employed to carry out a successful conservation translocation of the freshwater pearl mussel (*Margaritifera margaritifera*) in a declining population in northwest England. Following a small-scale pilot release in 2017, four release sites were identified for a population reinforcement of over 1300 tagged mussels in 2021. Monitoring during 2022 showed high levels of retention of juveniles at three out of the four release sites, despite the occurrence of a significant flood event during October 2021. Subsequent releases of 1100 juveniles were carried out across the three successful sites in 2023. Ongoing and regular monitoring is essential in order to provide data on the longer-term fate of propagated juveniles in the wild. This will allow for adaptive management of release activities in this river. These data will be useful to design conservation translocation strategies for other imperilled pearl mussel populations in the UK and throughout Europe.

**Keywords:** freshwater pearl mussel; *Margaritifera margaritifera*; population reinforcement; PIT tag; conservation translocation

## 1. Introduction

Freshwater mussels are among the most threatened taxa in Europe [1,2]. The population of freshwater pearl mussels (*Margaritifera margaritifera*) declined by over 90% during the 20th century [3] due to a wide range of factors, such as nutrient enrichment, siltation, and declines in host fish populations [3–6]. The species is classified as endangered on the IUCN Red List [7], and as such has been the subject of significant captive breeding activities across Europe over the last couple of decades [8] in efforts to save populations from extinction.

Captive breeding programmes face considerable challenges, such as the extended vulnerable juvenile period (approximately 12–15 years), water quality/breeding conditions, funding security/longevity, and the ability to recruit and retain experienced staff [8]. Despite these challenges, some programmes have started to release propagated juveniles/sub-adults in both population reinforcements (where individuals still exist in the wild) or reintroductions (where the population has become locally extinct), according to definitions from the IUCN Species Survival Commission [9].

Captive breeding strategies for freshwater mussels generally aim to rear a large number of juveniles to a size at which they stand a higher chance of surviving the sub-optimal habitat conditions currently still widespread in mussel rivers despite sometimes years of restoration efforts. The challenges in the wild are significant for juvenile *M. margaritifera*,

not least because the life cycle strategy involves an obligate parasitic stage on a salmonid fish [10], and juveniles must fall into clean yet stable habitats to continue growing [6]. Previous studies have shown that simply increasing the number of larvae (glochidia) on host fish in the wild is not enough to increase the population size if habitat conditions do not support high survival rates for the smallest juvenile stage [11]. Captive breeding therefore allows practitioners to maximise both the number of juvenile mussels produced and their survival rate during the most vulnerable life cycle stage. Large numbers of juveniles can be grown to sizes at which they are more robust before being released into the rivers from which their parent broodstock originated. Coupled with catchment restoration and/or habitat improvement initiatives, captive breeding and release projects have the potential to safeguard populations in the short term and ultimately save them from extinction.

Information and data on rearing systems and successful release methods are sparse and are not commonly published in the primary literature, leading to a paucity of information on successful release strategies, survival/growth post-release, and best-practice guidelines. Similarly, monitoring activities are commonly either too short-term, with the potential of reporting success prematurely [12], or completely absent [13], leading to uncertainty about the long-term outcomes of these efforts.

Here, we report on the successful population reinforcement and subsequent monitoring efforts of the freshwater pearl mussel population in the River Irt, Cumbria, UK. Once thought to number several hundreds of thousands of mussels, the population in the Irt now consists of only approximately 300 aging mussels. Only a single glochidium on one Atlantic salmon (*Salmo salar*) has been recorded over the past nine years, highlighting the critical condition of this population. Here, we outline population reinforcement activities in the Irt, including our site selection strategy, tagging and release methods, and water quality and hydrological parameters near release sites, as well as reporting monitoring results. It is hoped that sharing experiences of successful techniques in the Irt will be useful to other practitioners carrying out similar activities across Europe.

## 2. Materials and Methods

Population reinforcement activities took place in the River Irt, Cumbria, UK. The details of specific release sites are being kept confidential in order to protect against potential illegal poaching activities. Codes are used to identify individual sites. If required, further information can be obtained from the corresponding author.

### 2.1. Selection of Release Sites

The identification of potentially suitable release sites took place during 2019–2020. The historic distribution of mussels within the catchment was already known, but the prioritisation of potential release sites was necessary in order to target and monitor the success of releases.

Initially, the analysis of reach slope was carried out to identify sites where the slope was too steep, indicating that scouring would pose a high risk for mussels. The Detailed River Network data layer contained gradient data (m/km), which were interrogated with ArcGIS Desktop (10.8.2) to classify reaches as having a high likelihood of suitability (0–1.4 m/km), being potentially suitable (1.4–5 m/km), having a low likelihood of suitability (5–10 m/km), and displaying a very low likelihood of suitability (>10 m/km). A priority list of sites for further investigation was created based upon our knowledge of habitats and historic mussel distribution within the catchment, and the results of the slope analysis and rapid walkover surveys.

Priority sites were visited on one or more occasions during 2019 and 2020, and the extent of suitable juvenile mussel habitat and its condition was assessed using the method outlined by Killeen and Moorkens [14]. Briefly, during low-flow conditions (>Q85), we surveyed transects that spanned the river width every 10 m within the river reach. Biotic and abiotic parameters were recorded from 1 × 1 m quadrats across the entire transect, including parameters such as mussel number, substrate clast sizes, flow velocity, the presence

of algae/detritus, and silt infiltration. Quadrats were classified based on their suitability as juvenile mussel habitats (Good, Potential or No suitability) and their condition (Good, Moderate or Poor) [14]. These data were mapped using ArcGIS and four sites were short-listed for juvenile releases in 2021 (Table 1). Water quality sondes, measuring temperature (°C), conductivity (μS/cm), turbidity (NTU), chlorophyll-a (μg/L) and dissolved oxygen concentration (mg/L and %), logged data every 30 min at the PW release site and around 700 m upstream of the FG release site (site named RR) between July 2021 and December 2023. This RR site captured the input of a small tributary which delivers significant amounts of fine sediment to the Irt 700 m upstream of the FG release site. The effects of this tributary are diluted by the main river before the water reaches the mussel release site. Two additional short-term sondes were placed in BD and at the FG release site between February and April 2021 to compare water quality across the whole catchment. The water quality at BD and FG was relatively similar to that at other sites within the catchment, and so data collection was discontinued after April 2021.

**Table 1.** Description of shortlisted sites and their characteristics. Sites are listed from most upstream (BD) to most downstream (HB).

| Site Code | Historic/Current Mussel Presence? | Result of Slope Analysis | Notes on Site |
|---|---|---|---|
| BD | Anecdotal evidence of historic presence near this location | Potentially suitable | Newly reconnected historical side channel of main stem. Riparian habitat running through wet woodland, which protects channel from low and high flows through good hydrological and riparian connections. Varied sizes of substrate clasts with good depths of fine gravel. |
| PW | Currently low density on site | Potentially suitable | Site dominated by larger substrate clasts with lots of coarse gravel. Substrate very clean. Reach slope steeper than other sites. Woodland riparian habitat. River connection to floodplain limited. |
| FG | Currently low density but suspected high density area historically | Potentially suitable | Stable cobble layer overlying deep coarse sand and gravel substrates. Woodland riparian habitat, but river unable to connect with floodplain due to embankment and subsequent incised riverbed. Some minor bed armouring due to high flows being confined to the channel (unable to spread over floodplain). |
| HB | Currently low-density region, but suspected high-density area historically | Likely to be suitable | Stable cobble and pebble layer overlying coarse sand and gravel substrates. Patchy clay pockets interrupt areas of good mussel habitat. Woodland riparian habitat. Riverbed incised. but can connect with floodplain during highest flows. Some minor bed armouring due to high flows being confined to the channel. |

## 2.2. Captive Breeding and Tagging

The captive breeding of freshwater pearl mussels took place at the national breeding facility of the Freshwater Biological Association (FBA; Ambleside, Cumbria, UK). Briefly, broodstock mussels were translocated from the source river to the FBA facility in 2007 and used to produce several cohorts of juvenile mussels on an annual basis. Juveniles were collected as they excysted wild-strain hatchery-reared 0+ salmon (*Salmo salar*) and transferred to a range of breeding systems depending on their size and developmental stage (see [8] for a summary of the methods used). Upon reaching > 12 mm length, individuals were tagged with FPN 8 × 4 mm Hallprint tags (Hallprint, Hindmarsh Valley, Australia) and either Biomark HPT8 (8 mm) or APT12 (12 mm) Passive Integrated Transponder (PIT) tags (Biomark, St Boise, ID, USA). Hallprint tags were affixed to the left valve with Loctite (Henkel, Bridgewater, NJ, USA) cyanoacrylate glue. PIT tags were affixed to the right valve, firstly by sticking the tag to the shell with Loctite cyanoacrylate glue, and then by covering it with a thin layer of dental cement, as described in [15]. Juveniles were then

transferred to a flume until a suitable opportunity to translocate them to the river was identified. Juveniles which excysted from salmon in 2008, 2013, 2014 and 2017 were used for population reinforcements in the spring/summer of 2017 (pilot release [16], not considered here), 2021, and 2023 (see below). All individuals released were tagged to enable a robust monitoring programme post-release.

*2.3. Conservation Translocations and Monitoring*

All releases followed best-practice guidance for English freshwater pearl mussel translocations [17]. Periods of forecasted stable weather conditions were identified when river levels were suitable for safely undertaking juvenile freshwater mussel translocations and when flow velocity would not significantly vary for several days post-translocation. The releases took place during summer 2021 and 2023. Individuals from multiple cohorts were released on the same date, and more than one cohort was released into each site to protect against stochastic effects and potential genetic biases due to unequal broodstock contribution disproportionately impacting a particular cohort. Mussels were released into $1 \times 1$ m quadrats, which were identified as being either 'Good' or 'Potential' mussel habitats in 'Good' or 'Moderate' condition, as defined by Killen and Moorkens [14]. Preference was given to quadrats that were identified as 'Good' habitats in 'Good' condition. Five to twenty individuals were placed into each quadrat, and spaced out as much as possible to avoid signal interference from adjacent PIT tags, which could result in under-detection. Where possible, juveniles were placed in gravel underneath small cobbles to ensure that they were sufficiently buried and to reduce the risk of them becoming washed away shortly after placement. Where this was not possible, juveniles were placed in gravel deep enough so that their siphons were buried. Tag numbers and quadrat locations were recorded to aid future monitoring efforts.

Monitoring was carried out on two of the four release sites after the first releases in autumn 2021. All sites were subsequently monitored at least once during summer 2022 and at least twice during summer 2023. Monitoring was carried out with an HPR Plus Reader and BP Plus Antenna (Biomark, St Boise, ID, USA). Moving in an upstream direction, release sites were surveyed by sweeping near the riverbed with the antenna. Tag detections were automatically stored on the reader for subsequent download and analysis. Multiple passes were needed to cover the full channel width at all but one site. Data were downloaded with the Biomark Device Manager (v. 1.2.26), duplicates were removed, and data were copied to a master spreadsheet, including metadata on individual mussels and all tag detection history.

*2.4. Data Analysis*

The estimated retention at each site is reported for individuals released in 2021 only and is based upon monitoring data collected between 14 September 2021 and 6 September 2023. The retention of individuals from the 2023 releases is not considered here due to the low number of monitoring occasions after release.

## 3. Results

The site selection process identified steeper reaches, which could be discounted as release sites due to bed instability/scoured habitats unsuitable for mussels. Sites with a lower gradient were identified as potential/good suitability, but the gradient method also positively identified ponded reaches and low-flow areas unsuitable for juvenile mussels. This highlights the importance of local habitat knowledge when prioritising sites. After slope analysis, eleven sites were shortlisted for further investigation; four were subsequently identified as having enough Good or Potential mussel habitats in Good or Moderate condition to warrant carrying out population reinforcements at these sites. The main characteristics of each release site are detailed in Table 1, and examples of riparian and in-channel mussel habitat are highlighted in Figure 1. The details of releases, including

mean shell length, monitoring effort, and the estimated retention of juveniles per site, are provided in Table 2 based on monitoring data.

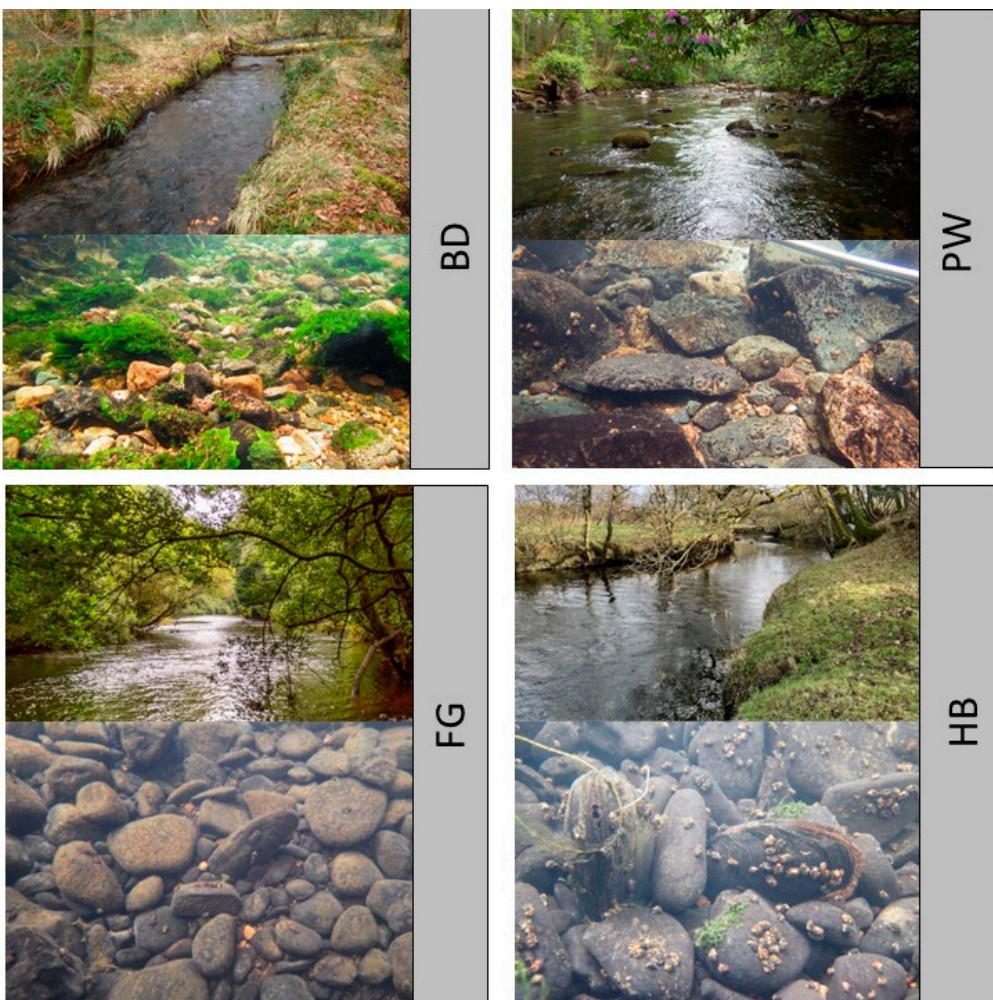

**Figure 1.** Photographs to illustrate riparian land coverage and substrate characteristics at each release site.

Juveniles were released when between 6 and 9 years old and when shell length was at least 15.0 mm, although the release of multiple age cohorts at the same time meant a high range of shell length at release in this study (15.0–70.7 mm).

Monitoring data showed that individuals could be missed on several monitoring occasions before being detected at a later date; indeed, every monitoring occasion resulted in several 'unique' recaptures which had not previously been detected. Immediately after individuals were placed at BD2 on 5 July 2023, the site was monitored for detection and only 70% of individuals were detected despite 100% being present. This highlights that monitoring results underestimate actual site occupancy in this river. Estimated retention at a site (Table 2) is given for individuals released in 2021 only and is based upon data from the most recent monitoring points. More monitoring occasions are required in order to develop quantitative modelling capable of predicting actual site occupancy.

Recapture rates were typically over 50% at most sites after two winters (Table 2) with the exception of PW, where low recapture rates (~13%) were consistently recorded after the first winter (2021/22). High winter flows (when juveniles are most vulnerable to being scoured out of the bed) peaked in October 2021 at 1.828 m at the closest gauging station (Galesyke), equating to a Q value of <1 (i.e., an event where the flow is exceeded <1% of the time) [18]. Flow gauging commenced on the River Irt in 1967, but peak flow data from prior

to 1978 are unreliable (Environment Agency pers. comm.). The daily peak flow observed in October 2021 has only been exceeded on three other occasions in October 1968, April 1975, and November 2009. Achieving good juvenile retention at three out of the four release sites, despite these significantly high winter flows, is promising.

**Table 2.** Details of juvenile mussel releases carried out between 2021 and 2023. Numbers of individuals released at each site are provided. These are split by age cohort (year juveniles excysted from fish). Mean shell length (mm) and range for each released cohort are provided, as well as the number of monitoring instances since the first release at that site. Estimated retention at site for individuals released in 2021 only, based upon repeat monitoring data. * Indicates that 70% were recorded on this monitoring occasion even though 100 % were present (monitoring carried out immediately after juvenile release).

| Site Code | Release Date | Number Released | Cohort | Mean Shell Length (Range) mm | No. Monitoring Instances | Estimated Retention at Site (%) |
|---|---|---|---|---|---|---|
| BD1 | 16 June 2021 | 49 | 2014 | 18.1 (16.5–20.3) | 4 | 82% |
| BD2 | 5 July 2023 | 76 | 2014 | 23.3 (17.0–30.0) | 1 | 70% * |
|  | 5 July 2023 | 23 | 2017 | 20.2 (14.0–25.0) |  |  |
| PW | 23 June 2021 | 47 | 2014 | 18.8 (15.3–21.0) | 4 | 13% |
|  | 15 July 2021 | 501 | 2014 | 14.6 (20.2–27.9) |  |  |
|  | 19 August 2021 | 27 | 2008 | 58.4 (44.7–65.3) |  |  |
|  | 19 August 2021 | 21 | 2013 | 28.6 (18.8–57.0) |  |  |
| FG | 23 June 2021 | 50 | 2014 | 19.7 (15.9–22.2) | 5 | 52% |
|  | 19 August 2021 | 29 | 2008 | 58.7 (46.8–70.7) |  |  |
|  | 19 August 2021 | 22 | 2013 | 27.5 (20.0–36.7) |  |  |
|  | 19 August 2021 | 244 | 2014 | 20.0 (15.0–27.1) |  |  |
|  | 11 July 2023 | 160 | 2014 | 24.0 (18.0–34.0) |  |  |
|  | 11 July 2023 | 38 | 2017 | 21.2 (17.0–26.0) |  |  |
|  | 10 August 2023 | 430 | 2014 | 25.6 (17.0–40.0) |  |  |
|  | 10 August 2023 | 76 | 2017 | 21.7 (15.0–29.0) |  |  |
|  | 6 September 2023 | 75 | 2014 | 25.6 (17.0–32.0) |  |  |
|  | 6 September 2023 | 5 | 2017 | 23.8 (20.0–27.0) |  |  |
| HB | 23 June 2021 | 48 | 2014 | 19.3 (15.4–21.9) | 3 | 60% |
|  | 18 August 2021 | 249 | 2014 | 20.2 (15.6–27.8) |  |  |
|  | 6 September 2023 | 182 | 2014 | 24.5 (16.0–35.0) |  |  |
|  | 6 September 2023 | 38 | 2017 | 20.9 (16.0–28.0) |  |  |
| Total |  | 2390 |  |  |  |  |

The monitoring parameters for water quality were generally within the expected ranges for viable freshwater pearl mussel populations (Table 3), as suggested in [19,20]. In winter, elevated turbidity events associated with high rainfall were generally brief, usually lasting for around 18 h. In June 2023, during a period of low flow, elevated temperatures around 20 °C were observed at PW for approximately 10 days, with a maximum of 22.2 °C. Lower-than-normal dissolved oxygen concentrations were observed during the same period at RR, dropping briefly to 7.09 mg/L (74.3%). Dead juveniles were only occasionally observed during the warmest and driest periods at PW, and so it was presumed that environmental conditions were largely suitable for good juvenile survival.

On two PIT monitoring occasions (one each at BD and PW), the substrate was excavated to confirm that PIT detections were of live mussels and not dead shells. All buried juveniles were found to be alive. Measurements of juveniles were only taken opportunistically when it was determined that this would not cause undue stress. All individuals had grown at both sites. At PW, individuals increased in size by between 32 and 72% of their original length after one year, and at BD individuals had grown by between 33 and 79%

of their original size two years after their release. Productivity was lower at BD (upper catchment) compared with PW (lower catchment); mean chlorophyll-*a* concentrations were 0.79 μg/L (±0.28 SD) at BD compared with 0.91 μg/L (±0.54 SD) at PW during February–April 2021. This difference in productivity could explain the slower growth rate observed at BD.

**Table 3.** Summary data for water quality sondes deployed at the two sites between February 2021 and December 2023.

| Sonde Site | Parameter | Mean (SD) | Min | Max |
|---|---|---|---|---|
| PW | Chlorophyll-a (μg/L) | 1.05 (0.8) | 0.01 | 11.67 |
| | Conductivity (μS/cm) | 2.11 (9.41) | 35.05 | 108.21 |
| | Dissolved oxygen (mg/L) | 11.18 (1.09) | 8.54 | 14.45 |
| | Dissolved oxygen (%) | 101.30 (3.37) | 91.80 | 118.56 |
| | Temperature (°C) | 11.28 (4.05) | 1.67 | 22.22 |
| | Turbidity (NTU) | 2.11 (3.4) | 0.01 | 87.52 |
| FG | Chlorophyll-a (μg/L) | 1.41 (0.95) | 0.02 | 20.68 |
| | Conductivity (μS/cm) | 93.32 (39.28) | 22.37 | 309.46 |
| | Dissolved oxygen (mg/L) | 10.87 (1.17) | 7.09 | 13.47 |
| | Dissolved oxygen (%) | 99.14 (5.42) | 74.34 | 125.65 |
| | Temperature (°C) | 11.55 (3.87) | 2.25 | 20.81 |
| | Turbidity (NTU) | 3.49 (5.50) | 0.67 | 197.19 |

## 4. Discussion

The number of freshwater mussel captive breeding programmes has increased substantially over the past 20 years [8]. However, reporting on the outcomes of reintroductions/population reinforcements remains limited in the primary literature. Accurate survival/site retention data are critical for informing adaptive management activities to ensure effective population restoration strategies [9,21]. This current paper outlines systematic methods undertaken to assess available juvenile mussel habitats (suitability and condition), identify candidate release sites, undertake population reinforcements with ~2500 propagated juveniles, and monitor the short-term success of releases.

This study details a successful population reinforcement that used older (larger) juveniles, which could be tagged and therefore monitored, to inform the success rate of releases. Whilst a small number of reproducing sub-adult mussels were released during this study (2008 cohort, unpublished data [22]), the majority of individuals released need to survive for a further ~3–5 years before they are large enough to contribute to the next generation. There are both benefits and costs associated with keeping juveniles in captivity for longer to release larger juveniles. On one hand, larger juveniles will be more resilient to sub-optimal habitat conditions and reach sexual maturity quicker than smaller juveniles. However, there are significant costs associated with keeping juveniles in captivity for extended periods and so the decision of when to release must be a compromise, considering factors such as available propagation resources and habitat quality. Previous studies have shown higher survival rates for larger juveniles/sub-adults [23,24], and population growth may be enhanced when sub-adults/younger adults are used due to higher reproductive value compared to juveniles, which have no immediate reproductive output [25]. Whilst it is possible for newly excysted juvenile mussels to survive in the cleanest substrates [26], many mussel rivers are currently not capable of supporting this most vulnerable life cycle stage [11], and monitoring is difficult due to small size. This study found that juveniles > 15 mm, whilst not sexually mature, survived well post-release and were relatively easy to monitor. Ongoing monitoring should be included as an essential activity in all release programmes in order to increase our knowledge of outcomes and the potential reasons for unforeseen issues [13,23,27–30].

The use of PIT tags for effective monitoring was critical in this study as most released juveniles were not visible at the surface. Previous studies on freshwater mussels highlighted their importance [15,31–33], but this study found that even with repeated monitoring, site

retention was likely to be under-estimated and that mussels might be missed on several monitoring occasions before being detected at a later date. One potential reason for this is signal disruption; if two or more PIT tags are positioned close together, this can lead to non-detection of one or more tags. At release densities of over 5 mussels/m² (this study 5–20 mussels/m²), signal disruption is probable. Secondly, in our study, juveniles were often placed under small cobbles in boulder/cobble-dominated habitats; large substrates like this can also interfere with signal detection. In fact, immediately after individuals were placed at BD2 on 5 July 2023, the site was monitored and only 70% of individuals were detected despite 100% being present. Published data on the retention of *M. margaritifera* after reintroductions/population reinforcements are patchy, but rates have been reported between 0 and 21% after 2–15 years [11]. Recapture rates of released mussels can vary significantly between sites and years. For example, the survival rate of the plain pocketbook (*Lampsilis cardium*) varied between 0 and 63% two years post-release across 13 different sites in three rivers [21]. This highlights the need for ongoing and repeated monitoring in order to refine population estimates.

Conservation translocations aim to establish a self-sustaining population that is genetically representative of the source [9,34]. Whilst not considered in this paper, genetic variation in juvenile cohorts reared for population reinforcements in the River Irt have been found to be genetically representative of the wild population [35]. The release of multiple cohorts across all sites and release dates is advised in order to protect against stochastic effects disproportionately impacting a particular cohort and any potential genetic bias which may arise due to unequal contributions from females in any one cohort [9,25,36–38].

Despite the lower retention rate seen during this study at the PW site, the other three release sites showed good survival and the limited growth data collected indicated that these sites were likely suitable for longer-term survival. The occurrence of a significant flood event in October 2021 appears to have had minimal impact on the retention of juveniles at three out of the four release sites based upon recapture data (Table 2). Scouring flows are suspected to have caused the downstream migration of a significant number of juveniles at PW, resulting in consistently lower numbers being recaptured at this site from spring 2022 onwards. Similar findings were reported in Poland, where a release site of the thick-shelled river mussel (*Unio crassus*) was found to be suitable only in low-flow years [12]. Further monitoring is required in the River Irt to ascertain whether these individuals were deposited downstream and whether they survived. A small number of live individuals were located just downstream of the release site (within 100 m), but individuals might have been transported several kilometres downstream [39], making accurate survival rates difficult to ascertain.

Environmental monitoring using logging sondes (Table 3) showed that water column water quality was mostly within the guidelines for expected favourable mussel habitats [19,20]. These data, however, do not tell us anything about the interstitial habitat conditions juvenile mussels occupy. The high survival and growth rates observed during this study however suggests that conditions are suitable for juveniles > 15 mm. Data showed that sub-optimal conditions caused by events such as long, dry-weather periods or high rainfall were generally short-term in duration and moderate in impact. Additionally, the River Irt is classified as achieving "Good Ecological Status" under the EU Water Framework Directive, with many of the biological and physico-chemical quality elements achieving "High Ecological Status" in 2019 (https://environment.data.gov.uk/catchment-planning/WaterBody/GB112074070070, accessed on 28 February 2024). The results of a pilot study using mussel silos (developed by Chris Barnhart, Missouri State University [40]) in the River Irt in 2020 and 2021 also gave us confidence that (water column) water quality was sufficient to support high juvenile survival and growth (unpublished data). An almost complete lack of wild-born sub-adults and young (smaller) adults in this population implies that interstitial habitat conditions are not yet capable of supporting newly excysted juveniles and that continued catchment-scale restoration is therefore needed alongside population reinforcements [8]. Future research and population reinforcements should

investigate whether smaller (<15 mm) and even newly excysted juveniles are capable of surviving in the interstitial habitat of the Irt.

The use of PIT tags to monitor mussel populations is a powerful tool and one which has enabled us to estimate retention per site two years after releases with relatively few monitoring occasions. The regular monitoring of release sites and downstream areas should be continued so that site-specific retention (and migration during high-flow events) can be modelled in the future [21,33,36,41] to help inform the adaptive management of this population reinforcement project.

## 5. Conclusions

Here, we highlight the successful population reinforcement activities of the freshwater pearl mussel in an English river, documenting our methods for site selection, release, and monitoring activities. The release of larger (>15 mm) juveniles is advantageous as they are less vulnerable to sub-optimal habitat conditions and their size allows for marking to aid post-release monitoring. A significant high-flow event shortly after the first releases in 2021 confirmed that three out of the four selected release sites were suitable for high rates of survival and juvenile retention. Lower retention was observed at the fourth site, indicating that substrates were too mobile and that bed stability is a critical consideration for release activities in future. Ongoing releases and monitoring will be key to demonstrating the longer-term success of these activities and to securing the future of this fragile mussel population.

**Author Contributions:** Conceptualization, L.L. and C.W.; methodology, L.L. and C.W.; software, L.L. and C.W.; formal analysis, L.L.; investigation, L.L. and C.W.; data curation, L.L. and C.W.; writing—original draft preparation, L.L. and C.W.; writing—review and editing, C.W.; visualization, L.L. All authors have read and agreed to the published version of the manuscript.

**Funding:** This research was funded by United Utilities through the River Ehen Habitats Regulations Compensatory Measures Package.

**Institutional Review Board Statement:** Not applicable.

**Data Availability Statement:** The original contributions presented in the study are included in the article, further inquiries can be directed to the corresponding author.

**Acknowledgments:** We thank the careful and considered thoughts of the two reviewers, whose contributions have improved the manuscript. We also thank our numerous colleagues from the Freshwater Biological Association and West Cumbria River Trust who provided support with fieldwork for population reinforcements and monitoring activities.

**Conflicts of Interest:** The authors declare no conflicts of interest.

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
