# Peer review of "Population Reinforcement of the Endangered Freshwater Pearl Mussel (Margaritifera margaritifera): Lessons Learned"

_diversity, doi:10.3390/d16030187_

Round 1
Reviewer 1 Report
Comments and Suggestions for Authors
This is an interesting and well-considered contribution to the study of using laboratory-reared freshwater mussels to bolster declining natural populations. The study contains much useful information on site selection, the composition of reinforcing mussels, how to place these in situ, and the interpretation of results.
I have a few minor comments on the manuscript, mostly referenced by line number.
(45) Not all freshwater mussel species use salmonid fish as hosts. This paragraph could be restricted to Margaritifera margaritifera but the problems which it lists are equally relevant to other species so I suggest omitting “salmonid”.
(292/293) Journal format for citation not used.
The “Author Contributions” and Acknowledgments” sections have not been adapted to the manuscript.
The titles of journal articles in the references are in title case (most words with initial capitals) rather than the Diversity format (Initial capital for the first word of the title and proper nouns).

The manuscript is very well written. There is however some mixing of U.K. and U.S. spelling (e.g. ‘imperiled” line 11, “programmes” line 34).
(133 & 135/136) “into” not “in to”
(152) “all but one” not including hyphens.
(185/186) Two instances of “only”. One should be removed.
(204) I suggest replacing “a high flow event” with “an event” which I think might make it easier to extrapolate to the meaning of, say, Q < 10.
(236) I suggest “ Only occasionally were dead juveniles observed …”
Table 3. The “a” in “Chlorophyll-a “ is not generally written in italic font (two instances)
(302) “do not” rather than “don’t”
Author Response
Thank you for your careful and considered thoughts on our manuscript - we are pleased you feel it makes a useful contribution to the body of knowledge on this under-reported topic.
I have made all of the changes you suggest, including elaborating on the need to monitor downstream of release sites to better quantify survival of wash-outs in lines 321-322.
Reviewer 2 Report
Comments and Suggestions for Authors
This study investigated the population reinforcement of the endangered freshwater pearl mussel (Margaritifera margaritifera). Introduction of the manuscript is correctly organized and attractive for the reader, cited literature is relevant to the topic. The experiments were in general well designed, and the results were also well presented. Therefore, I suggest a minor revision before acceptance for publication. My detailed comments are listed below.
Would you please introduce the annual production and commercial values of Margaritifera margaritifera?
What's the reasons for the Margaritifera margaritifera declining by over 90% during the 20th century?
It is not clear what's the meaning of the site code, could authors explain why these abbreviations used for site code.
It would be better to add a conclusion section in the end of the manuscript.
Please check the website of the endangered species on the IUCN Red List, the present link showed error.
Comments on the Quality of English Language
Minor editing of English language required.
Author Response
We thank the reviewer for their careful and considered thoughts on our manuscript.
On the question about annual production and commercial values; these are complicated questions which we don't believe are relevant to this paper. Our breeding programme produces tens of thousands of juveniles per year from several different populations, not all of which are relevant to this study. The actual survival rate to release is much lower than this (averaging around 20 - 50 % per year, depending upon the population), as is the case with most other freshwater mussel conservation programmes. This paper focusses on release activities rather than captive breeding statistics and we feel that trying to explain this would detract from the focus of the paper. This species doesn't hold any commercial value in the traditional sense, as it is not harvested or part of a commercial industry. It does however have it's own intrinsic value, but this is difficult to quantify in monetary terms. Because of these reasons, we haven't amended the text to include this information.
I have added some further information and references on the reasons for decline in the opening sentences; mainly due to habitat degradation caused by land use practices e.g. siltation, nutrient enrichment etc.
The site codes represent the initials of the actual site names. This is fairly common practice to keep the site codes similar to the actual site names so that if colleagues request further information or clarification, it is easy to reference and explain.
Short conclusion section added - thank you for the suggestion.
Amended IUCN Red List link - thank you for bringing this to our attention.
